# GeoVLMath: Enhancing Geometry Reasoning in Vision-Language Models via Cross-Modal Reward for Auxiliary Line Creation

## Abstract

Auxiliary lines are essential for solving complex geometric problems but remain challenging for large vision-language models (LVLMs). Rather than editing diagrams to draw auxiliary lines, which current image editing models struggle to render with geometric precision, we generate textual descriptions of auxiliary-line constructions to better align with the representational strengths of LVLMs. To bridge the gap between textual descriptions and spatial structure, we propose a reinforcement learning framework that enhances diagram-text alignment. At the core of our approach is a cross-modal reward that evaluates how well the generated auxiliary-line description for an original diagram matches a ground-truth auxiliary-line diagram. Built on this reward, we present GeoVLMath, an open-source LVLM tailored to auxiliary-line reasoning in solid geometry. This fine-grained signal drives a GRPO-based RL stage, yielding precise diagram-text alignment. To support training, we develop a scalable data creation pipeline and construct AuxSolidMath, a dataset of 3,018 real-exam geometry problems with paired diagrams and aligned textual fields. At the 3B and 7B scales, GeoVLMath achieves competitive and often superior performance compared with strong open-source and proprietary LVLMs on auxiliary-line reasoning benchmarks.

## 1 Introduction

Geometric problems constitute an important category of mathematical tasks, characterized by intricate spatial structures and multi-step reasoning processes (Ma et al., 2024). They are commonly divided into plane geometry and solid geometry. This study focuses on solid geometry, where reasoning over three-dimensional spatial relations is substantially more complex. Such problems rarely yield to direct application of standard theorems; instead, they often require the deliberate introduction of auxiliary lines[1] to reveal hidden geometric structure and enable further analysis. These constructions are essential for anchoring visual diagrams to formal symbolic reasoning and for providing the intermediate steps required for rigorous problem solving.

To validate the above idea, we conduct a pilot study with three preliminary experiments: one with incorrect auxiliary lines (Incorrect Aux), one without auxiliary lines (No Aux), and one with correct auxiliary lines (Correct Aux). As shown in Figure 1, **the use of correct auxiliary lines achieves the highest accuracy, whereas incorrect auxiliary lines lead to the poorest performance.** One possible explanation is that inaccurate auxiliary lines tend to misdirect reasoning and produce errors, while precise auxiliary lines uncover key spatial relationships, thereby enhancing solution accuracy.

Given that accurate auxiliary lines are crucial for solution correctness, the key question is how to obtain them reliably. The most straightforward approach is to draw auxiliary lines directly on the original diagram. However, current image editing models cannot accurately draw auxiliary lines (as illustrated in Figure 5). Another representative approach adopts a tool-use pipeline, exemplified by Visual Sketchpad (Hu et al., 2024): an LVLM is prompted to generate code that draws auxiliary lines, which is then rendered into an augmented diagram. Although effective in controlled settings, this method faces two key limitations. First, it depends on the specific coordinate position of the diagram elements, which is rarely available in real-world problems.

---

[1]In this paper, we use *auxiliary lines* broadly to include both additional lines and coordinate systems.

Additionally, it relies on the LVLM's ability to generate highly accurate code for auxiliary lines. This dependency constrains its robustness and limits its applicability to broader classes of geometric problems. Taken together, these limitations indicate that neither direct image editing nor coordinate-dependent tool usage approaches can reliably produce accurate auxiliary lines in current technology, motivating an approach that avoids image manipulation and strict coordinate requirements.

Motivated by the success of textual chain-of-thought, we avoid image editing and explicit coordinates by expressing auxiliary-line constructions in text and verifying them against the diagram. Our

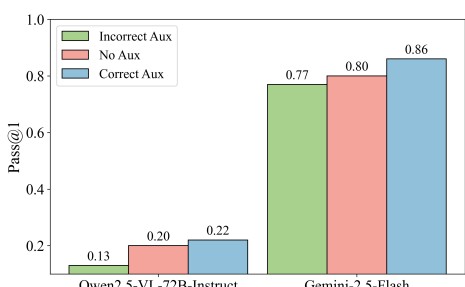

Figure 1: Pass@1 of Qwen2.5-VL-72B-Instruct and Gemini-2.5-Flash. "Aux" denotes auxiliary-line description.

main idea is to design a **cross-modal reward model** that measures the consistency between a generated textual auxiliary-line description for the original diagram and a ground-truth auxiliary-line diagram. The reward is computed by jointly encoding the original diagram and the generated auxiliary-line description, and then comparing this pair with the ground-truth auxiliary-line diagram, providing geometry-aware supervision without requiring coordinate assumptions or image manipulation. Building on this reward signal, we train a policy model using Group Relative Policy Optimization (GRPO) to obtain geometry-consistent, generalizable constructions. Training follows a two-stage paradigm inspired by recent progress in reinforcement learning (RL) for reasoning (e.g., DeepSeek-R1 (Guo et al., 2025)): supervised fine tuning (SFT) for cold start, followed by GRPO-based RL to further elicit structured reasoning and strengthen diagram-text alignment. We instantiate the framework as GeoVLMath (3B/7B), a vision-language model tailored to auxiliary-line-based geometric reasoning. Through the cross-modal supervision, GeoVLMath achieves strong alignment between text and geometric structure, enabling faithful reasoning on complex diagrams.

To effectively train the above model, we require a high-quality dataset that captures both visual and symbolic aspects of real-world geometry problems. Yet, creating such a dataset is inherently challenging due to the need for automation, scalability, and semantic precision across diverse and noisy educational materials. In response to these challenges, we develop a **robust and scalable data construction pipeline** specifically designed to transform raw high school exam papers into structured multimodal instances suitable for training LVLMs. (1) First, we develop automated scripts to identify and filter geometry problems that satisfy predefined criteria from a large corpus of exam papers; (2) Next, we apply a fully automated pipeline to deduplicate problems and extract their associated geometric diagrams, which are filtered to remove low-resolution or unclear diagrams and then saved for subsequent processing; (3) Subsequently, we train a specialized text extraction model capable of handling MathType equations. This model accurately parses the problem description, the final answer, and the auxiliary-line description from exam papers and converts them into a structured JSON format; (4) Finally, we perform a manual verification step to ensure the accuracy, completeness, uniqueness, and semantic consistency of each data instance. While the pipeline is largely automated, this lightweight manual verification step is essential for maintaining data quality, particularly when handling complex symbolic expressions and diagrammatic content in real-world settings. Based on this pipeline, we construct **AuxSolidMath**, a curated dataset of 3,018 solid geometry problems in a rich multimodal format, comprising the problem description, the final answer, the auxiliary-line description, the original diagram, and the auxiliary-line diagram. To our knowledge, AuxSolidMath is the first systematically constructed dataset explicitly tailored to auxiliary-line-based solid geometry reasoning. Beyond this specific resource, our pipeline offers a scalable framework that can be readily adapted to generate large-scale, high-quality datasets for multimodal reasoning tasks.

To assess the effectiveness of our model, GeoVLMath, we conduct a comprehensive evaluation against frontier open-source and closed-source LVLMs. Despite its relatively modest parameter scale, GeoVLMath achieves highly competitive performance, frequently outperforming substantially larger models such as Qwen2.5-VL-32B-Instruct (Bai et al., 2025) and GPT-4o (Hurst et al., 2024). These results suggest that supervision grounded in auxiliary-line constructions is more effective for enhancing geometric reasoning than simply scaling model parameters. Furthermore, our evaluation protocol highlights the utility of datasets augmented with auxiliary lines in revealing the

limitations of existing LVLMs in structured visual reasoning, and provides a principled foundation for future model design and evaluation.

The main contributions of this work are threefold:

- **Cross-modal reward for auxiliary-line alignment.** We propose a geometry-aware scalar reward that contrasts a fused diagram-text representation with the ground-truth auxiliary-line diagram, requiring no coordinates or image edits. Training with GRPO turns this signal into effective RL supervision, improving diagram-text alignment and spatial reasoning.

- **AuxSolidMath: a dataset for auxiliary-line geometric reasoning.** We design a scalable, modular pipeline to construct AuxSolidMath, a dataset tailored to auxiliary-line reasoning in solid geometry problems. Curated from authentic high-school exams, it comprises 3,018 richly annotated multimodal instances pairing diagrams with aligned textual fields, providing training data for our proposed framework. AuxSolidMath will be publicly released on Hugging Face to enable community use and further research.

- **GeoVLMath(3B/7B): Competitive Small-Scale Models for Geometric Reasoning.** We introduce GeoVLMath, a LVLM trained on AuxSolidMath and optimized with a GRPO-based RL framework. Despite its relatively modest parameter scale, GeoVLMath achieves performance comparable to, or even surpassing, that of substantially larger models such as Qwen2.5-VL-32B-Instruct and GPT-4o on geometric reasoning tasks requiring auxiliary line constructions.

## 2 METHODOLOGY

### 2.1 PROBLEM DEFINITION

We study geometry problems whose solution requires the *active construction of auxiliary lines*. Formally, each instance is represented as a pair $\langle I, q \rangle$, where $I$ denotes an original diagram, and $q$ is a question grounded in $I$. The goal is to generate a solution $y$ that includes the auxiliary lines $aux$, a sequence of deductive steps, and a final answer $ans$. Since auxiliary lines are absent from the original diagram $I$, they must be constructed during reasoning. The deliberate introduction of these lines uncovers latent spatial relations and formalizes them as explicit geometric constraints, thereby enabling solutions that would otherwise be difficult to derive from the original diagram alone.

### 2.2 FRAMEWORK OVERVIEW

We introduce a two-stage training framework for LVLMs that integrates the auxiliary-line construction into the reasoning process. In the first stage, we apply supervised fine-tuning (SFT) on automatically synthesized chain-of-thought (CoT) data with explicit auxiliary-line steps, enabling the model to actively construct auxiliary lines, thereby establishing a good initialization. In the second stage, we further use reinforcement learning to encourage the model to construct auxiliary lines that faithfully reflect the geometry of the diagram, boosting the precision of the solution. At the core is a **cross-modal reward model** that provides fine-grained feedback by scoring the agreement between the original diagram plus the generated auxiliary-line description and a reference diagram annotated with the correct auxiliary line. Overall, this framework combines direct supervision with structured visual feedback, resulting in more reliable auxiliary-line constructions and stronger geometric problem-solving performance.

### 2.3 SUPERVISED FINE-TUNING FOR COLD-START INITIALIZATION

We start with SFT on COT exemplars to provide a cold-start initialization for subsequent reinforcement learning. Each training instance is represented as a triplet $\langle I, q, y \rangle$, where $I$ denotes an original geometric diagram, $q$ is a natural language question, and $y = (y_1, ..., y_T)$ is a step-by-step solution that includes the construction of necessary auxiliary lines $aux$, intermediate deductions, and the final answer $ans$. To facilitate structured supervision and prepare for reward modeling in the RL stage, we mark auxiliary lines with special tokens [AUX] and [/AUX] within $y$. These markers expose auxiliary-line steps as explicit supervision signals. We fine-tune the model using a standard next-token prediction objective. The SFT loss is defined as:

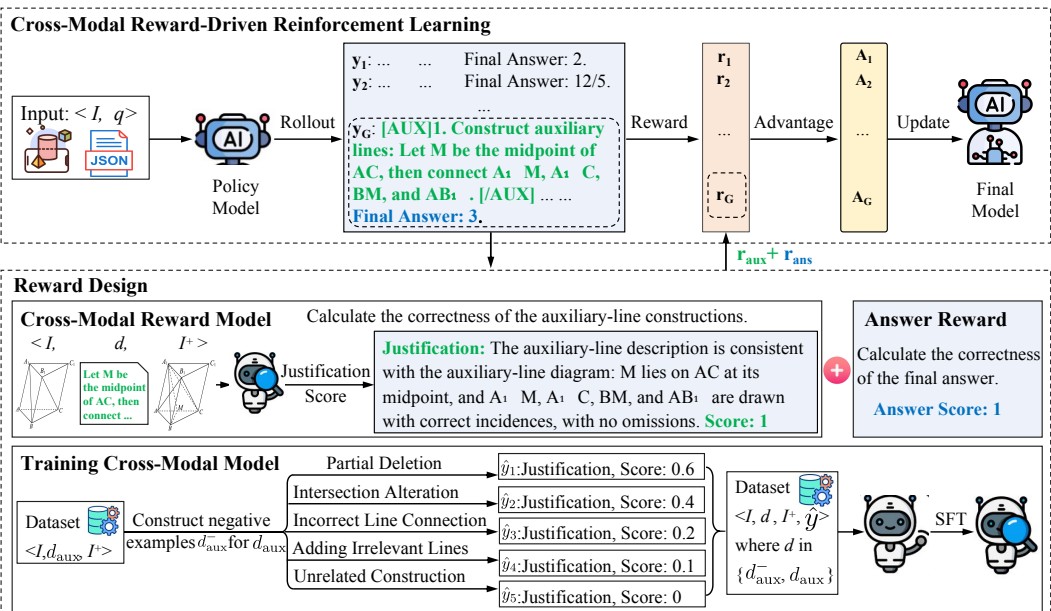

Figure 2: Overview of the cross-modal reward-driven RL. We first fine-tune a cross-modal reward model on curated high-quality data to evaluate the correctness of auxiliary-line constructions. During the RL phase, the reward model's consistency score is combined with a final-answer accuracy reward to produce a composite signal that updates the policy via GRPO.

$$\mathcal{L}_{\text{SFT}} = -\mathbb{E}_{(I,q,y)\sim\mathcal{D}} \left[ \sum_{t}^{T} \log P_\theta(y_t \mid I, q, y_{<t}) \right] \tag{1}$$

where $\mathcal{D}$ is the training set, $\theta$ denotes the model parameters, $y_t$ indicates the $t$-th output token, and $y_{<t}$ comprises all previously generated output tokens.

## 2.4 CROSS-MODAL REWARD-DRIVEN REINFORCEMENT LEARNING

As shown in Figure 1, accurate auxiliary-line constructions improve reasoning success. The key challenge is how to integrate these constructions into the reasoning process. The most intuitive approach is to draw auxiliary lines via image editing models or adopt coordinate-dependent pipelines, but limited editing reliability and the need for precise, accessible coordinates often render both impractical. Motivated by the recent advances of textual COT supervision, we represent auxiliary-line constructions in natural language and propose a **cross-modal reward model** that scores diagram-text alignment between the original diagram together with the textual description and a reference diagram annotated with the ground-truth auxiliary lines. This resulting signal is geometry-aware yet agnostic to image editing and strict coordinate assumptions, enabling scalability across diverse diagram styles. We integrate this cross-modal reward, alongside a final-answer reward, into a GRPO-based RL stage to align intermediate constructions with the diagram while maintaining final-answer accuracy. An overview of the RL stage is illustrated in Figure 2.

### 2.4.1 CROSS-MODAL REWARD MODEL FOR DIAGRAM-TEXT ALIGNMENT

Given an original diagram $I$, a textual description $d$ of auxiliary lines (either the correct $d_{\text{aux}}$ or a perturbed $d_{\text{aux}}^-$), and a reference diagram $I^+$ annotated with the ground-truth auxiliary lines, the reward model compares the *relations induced by applying $d$ to $I$* against the *additional geometric structures present in $I^+$ but absent from $I$*. Beyond surface similarity, it evaluates diagram-text relational consistency, such as parallelism, orthogonality, and angle bisection, and scores whether these relations are satisfied in the reference diagram. Accordingly, the reward exhibits three desirable properties: **(1) cross-modal relational alignment**, measuring diagram-text consistency at the level of geometric relations, rather than lexical similarity; **(2) Coordinate-free and render-free**, requiring neither

pixel-accurate coordinates nor explicitly drawing the specified auxiliary lines on the diagram, since the alignment is computed from diagram-text correspondence; and **(3) Fine-grained**, providing intermediate scores for partially correct yet meaningful constructions, thereby enabling precise credit assignment across multi-step reasoning. In summary, the cross-modal reward precisely measures diagram-text spatial consistency, capturing whether and to what extent the generated auxiliary lines match the intended geometric structure, without explicitly rendering on the diagram or relying on coordinate-dependent methods. Next, we explain how to **construct diagram-text supervision** automatically and at scale, and **train the cross-modal reward model** accordingly.

**Constructing Diagram–Text Supervision.** Each training example is represented as $\langle I, d, I^+, \hat{y} \rangle$, where $\hat{y} = (r, s)$ contains a brief justification $r$ and a consistency score $s \in [0, 1]$ indicating how well $d$ aligns with $I^+$ given $I$. We construct this supervision dataset via a fully automated pipeline (see Figure 2). Starting from high-quality triplets $\langle I, d_{\text{aux}}, I^+ \rangle$ sourced from the dataset described in Section 3, we introduce rule-based perturbations that simulate common errors, including *partial deletion, intersection alteration, incorrect line connections, adding irrelevant lines, unrelated auxiliary lines*. Based on these templates, we then leverage a large language model to generate diverse and linguistically varied negatives $d_{\text{aux}}^-$ that are lexically plausible yet geometrically inconsistent with the intended construction in the reference diagram. To evaluate the consistency between each description $d \in \{d_{\text{aux}}, d_{\text{aux}}^-\}$ and its target construction $I^+$, we employ an *LVLM-as-a-Judge* strategy. Specifically, the LVLM is prompted to assess diagram-text alignment for the pair $\langle I, d \rangle$ against $I^+$ and to output both a natural language rationale $r$ and a scalar score $s \in [0, 1]$. This automated evaluation provides interpretable justifications and continuous alignment signals, enabling scalable, fine-grained supervision that spans faithful descriptions through adversarial counterexamples.

**Training the Cross-Modal Reward Model.** Given the input triplet $\langle I, d, I^+ \rangle$, the model outputs a structured prediction $\hat{y} = (r, s)$, where $r$ is a rationale and $s \in [0, 1]$ is a consistency score. We train the model by maximizing the conditional likelihood of the serialized output:

$$p_\phi(\hat{y} \mid I, d, I^+) = \prod_{i=1}^{T} p_\phi(\hat{y}_i \mid I, d, I^+, \hat{y}_{<i}) \tag{2}$$

where $T$ denotes the length of the generated sequence $\hat{y}$, $\hat{y}_i$ is the $i$-th token in the output, and $\hat{y}_{<i}$ represents the sequence of previously generated tokens. The consistency score is indicated as $r_{\text{aux}} = s = \text{Score}(\hat{y})$, where higher values correspond to better consistency.

Through this training, we obtain a reward model that provides precise, interpretable feedback on diagram-text alignment between auxiliary-line descriptions and the reference diagram. This model serves as a key component of our RL framework, guiding the policy toward auxiliary-line constructions that are geometrically consistent and diagram-grounded.

### 2.4.2 OPTIMIZATION

We adopt GRPO as the policy optimization algorithm. The overall reward signal combines the cross-modal reward introduced in Section 2.4.1 with a final-answer reward defined as a binary score, yielding 1 if the predicted final answer matches the ground truth and 0 otherwise, *i.e.*,

$$r = \alpha r_{\text{aux}} + (1 - \alpha) r_{\text{ans}} \tag{3}$$

Given a geometric diagram $I$ and a question $q$, GRPO samples a set of response sequences $\{y_1, y_2, \ldots, y_G\}$ from the old policy model $\pi_{\theta_{\text{old}}}$. The policy model $\pi_\theta$ is then optimized by maximizing the following objective, following the formulation introduced in (Guo et al., 2025):

$$\mathcal{L}_{\text{GRPO}} = \frac{1}{G} \sum_{i=1}^{G} \left( \min \left( \frac{\pi_\theta(y_i \mid I, q)}{\pi_{\theta_{\text{old}}}(y_i \mid I, q)} A_i, \, \text{clip} \left( \frac{\pi_\theta(y_i \mid I, q)}{\pi_{\theta_{\text{old}}}(y_i \mid I, q)}, \, 1 - \epsilon, \, 1 + \epsilon \right) A_i \right) - \beta \, \mathbb{D}_{\text{KL}}(\pi_\theta \parallel \pi_{\text{ref}}) \right) \tag{4}$$

Here, $G$ denotes the group size, while $\epsilon$ and $\beta$ are hyperparameters for clipping and KL penalty.

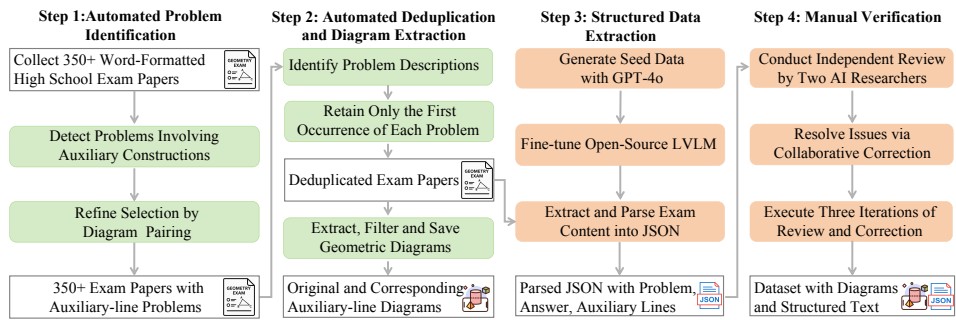

Figure 3: Overview of the Proposed Data Creation Pipeline.

## 3  DATA CREATION

Building on this framework, we curate the AuxSolidMath dataset to support model training. As illustrated in Figure 3, our data creation pipeline proceeds through four progressive steps: **automated problem identification, automated deduplication and diagram extraction, structured data extraction, and manual verification**. The pipeline standardizes raw exam problems into semantically aligned, high-quality multimodal instances that support training vision-language models for auxiliary-line geometric reasoning (More details in Appendix D). Figure 4 shows an example from the dataset. Each instance is represented as a five-tuple consisting of the *problem description*, *the final answer*, *the auxiliary-line description*, *the original diagram*, and *the auxiliary-line diagram*.

**Automated Problem Identification.** We curate AuxSolidMath from 350+ authentic high-school geometry exam sets by automatically selecting problems that explicitly require auxiliary-line constructions. A two-stage filter consists of cue-verb retrieval in reference solutions (e.g., "connect," "construct," "draw," "establish") and verification of paired diagrams (original and auxiliary-line-annotated), retaining only items that meet both criteria.

**Automated Deduplication and Diagram Extraction.** We automatically deduplicate the dataset and extract paired diagrams to ensure unique and high-quality instances. **Problem Deduplication.** Duplicates are detected by textual matching, and only the first occurrence is retained. **Diagram Extraction.** For each retained problem, we extract the original and auxiliary-line-annotated diagrams and apply OpenCV-based filtering to discard low-resolution or unclear images.

**Structured Data Extraction.** Building on the high-quality diagram pairs obtained in the previous step, we extract three textual fields for each problem: the problem description, the final answer, and the auxiliary-line description. Because the source Word files embed MathType formulas that standard parsers handle poorly, we render pages as images and parse them with a vision-language model. To ensure accuracy, we curate a 300-example seed verified by GPT-4o (Hurst et al., 2024) and use it to fine-tune Qwen2.5-VL-7B-Instruct (Bai et al., 2025), yielding a reliable, domain-adapted extractor. All outputs follow a unified JSON schema for downstream training.

**Manual Verification.** Each instance is independently reviewed by two AI researchers for accuracy, completeness, uniqueness, semantic consistency, as well as visual clarity and resolution. If either reviewer flags an issue, the instance is collaboratively revised, with up to three rounds per instance to systematically identify and correct even subtle or ambiguous errors. A human review resolves complex symbolic expressions and ambiguous diagrammatic elements that are often misinterpreted by automated tools. Despite its modest cost and effort, this step remains indispensable for maintaining the high data fidelity required for reliable model training and evaluation.

**Dataset Statistics.** Our dataset AuxSolidMath comprises 3,018 solid geometry problems collected from real high school examination papers. Within this dataset, we curate a new benchmark, GeoAuxBench, designed specifically to evaluate a model's ability to construct auxiliary lines, a skill essential to solving complex geometry problems. GeoAuxBench contains 302 examples and is divided into two difficulty levels, *Easy* (150) and *Hard* (152), using the original difficulty annotations from the source exam papers rather than post hoc labels. The tiers reflect increasing reasoning complexity along three axes: (i) the sophistication of auxiliary-line constructions, (ii) the depth of multi-step reasoning, and (iii) the degree of implicit spatial inference. This design provides a principled basis for evaluating models across different levels of geometric reasoning difficulty.

**Question:** Given a cube ABCD - $A_1$ $B_1$ $C_1$ $D_1$ with edge length 1, points P and Q are moving points on edges $C_1$ $D_1$ and $B_1$ C, respectively. Determine the maximum volume of the tetrahedron PQAD.

**Final Answer:** 1/6

**Auxiliary-line Description:** Through point Q, draw QG parallel to $B_1$ $C_1$ intersecting $CC_1$ at G. Connect PG, GD, DP, and AP.

Original Diagram          Auxiliary-line Diagram

Figure 4: An Example from the AuxSolidMath Dataset.

# 4 EXPERIMENTS

## 4.1 EXPERIMENTAL SETTINGS

**Benchmark.** We evaluate on GeoAuxBench, a benchmark subset of AuxSolidMath designed to evaluate auxiliary-line constructions. Existing benchmarks largely target general geometric reasoning and seldom require the introduction of auxiliary lines, making them misaligned with our task, especially in solid geometry. GeoAuxBench spans two difficulty tiers, **Easy** and **Hard**, providing a comprehensive testbed for evaluating the geometric reasoning capabilities of LVLMs.

**Metrics.** To evaluate model performance, we adopt the Pass@k (Chen et al., 2021b) metric, a widely used evaluation criterion originally introduced by OpenAI for assessing solution correctness under sampling. We report Pass@1 and Pass@5: Pass@1 is the accuracy of a single sample, and Pass@5 is the proportion of problems for which at least one of five samples is correct.

**Models.** We evaluate GeoVLMath at two model sizes: 3B and 7B, all built on the Qwen2.5-VL backbone (Qwen2.5-VL-3B/7B) (Bai et al., 2025). To comprehensively assess its performance, we benchmark it against 18 state-of-the-art LVLMs, encompassing both closed-source and open-source models. Further implementation details are shown in Appendix E.

## 4.2 MAIN RESULTS

We evaluate GeoVLMath against 18 baseline models on the GeoAuxBench benchmark, with results presented in Table 1. We summarize three key findings: **(1) GeoVLMath consistently demonstrates superior performance on GeoAuxBench.** GeoVLMath-3B and GeoVLMath-7B outperform their base models, Qwen2.5-VL-3B-Instruct and Qwen2.5-VL-7B-Instruct, on pass@5. Specifically, GeoVLMath-3B achieves an absolute gain of **+3.55%** (from 11.61% to 15.16%), and GeoVLMath-7B improves by **+10.19%** (from 15.93% to 26.12%). We attribute these gains to the auxiliary-line-aware training signal. During training, a cross-modal reward model evaluates the generated auxiliary-line description against a reference auxiliary-line diagram and returns a diagram-text alignment score. This supervision guides the model to recognize when auxiliary lines are warranted, position them in appropriate locations, and exploit the resulting constraints to complete multi-step derivations. In contrast, baseline models lack this targeted alignment and typically do not proactively construct auxiliary lines, instead relying on direct theorem application and consequently overlooking latent geometric structure. As a result, GeoVLMath exhibits stronger diagram-text grounding, clearer reasoning, and more reliable solutions. **(2) GeoAuxBench-Hard is a challenging benchmark that clearly distinguishes LVLM capabilities in auxiliary-line-aware geometric reasoning.** On the pass@1 metric, GeoAuxBench-Hard yields a clear separation in performance: the leading models, Gemini-2.5-Flash and gpt-5-mini, attain only **63.16%**, whereas the open-source Qwen2.5-VL-72B-Instruct reaches **13.16%**. By design, the benchmark demands deliberate auxiliary-line construction and multi-step spatial reasoning while minimizing shortcut opportunities, and it provides reference diagrams for fine-grained error analysis. These attributes establish GeoAuxBench-Hard as a concise yet highly discriminative testbed for reliably distinguishing LVLM capabilities. **(3) Model scale alone does not compensate for insufficient auxiliary-line awareness.** On GeoAuxBench-Easy, GeoVLMath-7B achieves higher pass@5 than Qwen2.5-VL-32B-Instruct (**35.56%** vs. **23.33%**). On GeoAuxBench-Hard, GeoVLMath-7B also outperforms Qwen2.5-VL-32B-Instruct (**16.67%** vs. **13.16%**). Error analysis indicates that the Qwen2.5-VL-32B-Instruct rarely constructs auxiliary lines, thereby overlooking latent spatial constraints. Instead, GeoVLMath-7B more proactively introduces auxiliary lines in appropriate locations and exploits the

Table 1: Overall evaluation on GeoAuxBench (%).

| LVLM | Easy | | Hard | | Average | |
|---|---|---|---|---|---|---|
| | Pass@1 | Pass@5 | Pass@1 | Pass@5 | Pass@1 | Pass@5 |
| **Closed-source LVLMs** | | | | | | |
| gpt-5-mini | 89.33 | 92.67 | 63.16 | 75.00 | 76.25 | 83.84 |
| o4-mini-2025-04-16 | 84.00 | 93.33 | 60.53 | 74.34 | 72.27 | 83.84 |
| GPT-4o | 8.67 | 25.33 | 6.58 | 15.13 | 7.63 | 20.23 |
| Gemini-2.0-Flash | 37.33 | 62.67 | 25.00 | 39.47 | 31.17 | 51.07 |
| Gemini-2.5-Flash | 84.00 | 91.33 | 63.16 | 78.95 | 73.58 | 85.14 |
| Claude 3.7 Sonnet 20250219 | 15.33 | 41.33 | 13.16 | 28.29 | 14.25 | 34.81 |
| Claude Sonnet 4 20250514 | 56.00 | 77.33 | 30.92 | 44.74 | 43.46 | 61.04 |
| **Open-source LVLMs (3B-14B)** | | | | | | |
| InternVL3-8B | 9.33 | 25.33 | 5.92 | 15.79 | 7.63 | 20.56 |
| Llama-3.2-11B-Vision-Instruct | 2.00 | 12.00 | 3.29 | 5.92 | 2.65 | 8.96 |
| InternVL3-14B | 13.33 | 28.67 | 5.92 | 15.13 | 9.63 | 21.90 |
| Qwen2.5-VL-3B-Instruct | 2.00 | 14.89 | 1.97 | 8.33 | 1.99 | 11.61 |
| **GeoVLMath-3B (Ours)** | 12.89 | 20.44 | 5.70 | 9.87 | 9.30 | 15.16 |
| Qwen2.5-VL-7B-Instruct | 5.14 | 20.67 | 3.95 | 11.18 | 4.55 | 15.93 |
| **GeoVLMath-7B (Ours)** | **14.67** | **35.56** | **5.92** | **16.67** | **10.30** | **26.12** |
| **Open-source LVLMs (17B-78B)** | | | | | | |
| Qwen2-VL-72B-Instruct | 6.00 | 15.33 | 5.26 | 8.55 | 5.63 | 11.94 |
| Qwen2.5-VL-32B-Instruct | 20.67 | 23.33 | 11.18 | 13.16 | 15.93 | 18.25 |
| Llama-4-Scout-17B-16E-Instruct | 20.67 | 36.67 | 7.89 | 18.42 | 14.28 | 27.55 |
| InternVL3-38B | 19.33 | 41.33 | 10.53 | 21.71 | 14.93 | 31.52 |
| Qwen2.5-VL-72B-Instruct | 24.00 | 40.67 | 13.16 | 19.74 | 18.58 | 30.21 |
| InternVL3-78B | 16.67 | 36.67 | 9.21 | 21.05 | 12.94 | 28.86 |

[*] Bold indicates the best results for models of similar sizes.

resulting constraints in subsequent steps. These results indicate that supervision of the auxiliary line, rather than parameter count alone, is the decisive factor for reliable reasoning.

### 4.3 CROSS-MODAL REWARD MODEL

Leveraging AuxSolidMath triplets $\langle I, d_{\text{aux}}, I^+ \rangle$, we apply rule-based perturbations to simulate typical auxiliary-line errors and use the resulting data to train a cross-modal reward model on Qwen2.5-VL-7B. The dataset comprises 2,970 training examples and 330 test examples. We train the model for 3 epochs with a batch size of 16, using the AdamW optimizer with a learning rate of 2e-5 and a cosine learning rate scheduler with a 0.1 warm-up ratio. During training, the vision tower and projection modules are frozen, while the language model remains trainable. The model achieves a pass@1 accuracy of **98.18%** on the test set, indicating reliable alignment between textual auxiliary-line descriptions and their visually annotated counterparts. This result further validates cross-modal supervision as an effective means of producing geometry-aware rewards, providing signals that assess whether the proposed constructions satisfy the underlying geometric constraints of the diagram without resorting to image manipulation or explicit coordinates.

### 4.4 ABLATION STUDIES

**Cross-Modal Reward.** We assess the role of cross-modal supervision with two variants, keeping all other settings unchanged. (a) *w/o cross-modal reward.* This variant removes all supervision related to auxiliary lines and trains the model solely for final-answer accuracy, with no supervision on whether auxiliary lines are introduced. This isolates the effect of answer-only supervision and approximates a setting where auxiliary lines are omitted from the training objective. (b) *Textual reward.* Cross-modal consistency is replaced by a text-only semantic similarity objective that measures the proximity between the generated auxiliary-line description and the ground-truth annotation. Concretely, we use *EmbeddingGemma* (DeepMind, 2025c) to encode sentences and compute a similarity score for training. This variant favors fluent textual descriptions but does not enforce grounding to the input diagram. *Findings.* As reported in Table 2, removing the cross-modal reward results in performance degradation (Average pass@1: **10.30** → **7.86**, pass@5: **26.12** → **22.91**), underscoring the importance of geometry-aware supervision for instances that require introducing auxiliary lines. Substituting it with a purely textual similarity objective performs even worse (Average pass@1: **10.30** → **7.53**, pass@5: **26.12** → **20.47**), consistent with our pilot finding in Section 1 that incorrect auxiliary lines can be worse than none. These declines suggest that lexical alignment introduces

Table 2: Results of ablation studies (%).

| | Easy | | Hard | | Average | |
|---|---|---|---|---|---|---|
| | **Pass@1** | **Pass@5** | **Pass@1** | **Pass@5** | **Pass@1** | **Pass@5** |
| GeoVLMath-7B | 14.67 | 35.56 | 5.92 | 16.67 | 10.30 | 26.12 |
| w/o Cross-Modal Reward | $10.89_{\downarrow3.78}$ | $32.22_{\downarrow3.34}$ | $4.82_{\downarrow1.10}$ | $13.60_{\downarrow3.07}$ | $7.86_{\downarrow2.44}$ | $22.91_{\downarrow3.21}$ |
| Textual Reward | $10.67_{\downarrow4.00}$ | $28.44_{\downarrow7.12}$ | $4.39_{\downarrow1.53}$ | $12.50_{\downarrow4.17}$ | $7.53_{\downarrow2.77}$ | $20.47_{\downarrow5.65}$ |
| w/o RL | $3.33_{\downarrow11.34}$ | $20.44_{\downarrow15.12}$ | $3.95_{\downarrow1.97}$ | $11.18_{\downarrow5.49}$ | $3.64_{\downarrow6.66}$ | $15.81_{\downarrow10.31}$ |

spurious signals and conflicts with precise diagram grounding, favoring surface-level paraphrases over geometry-aware reasoning. Error analysis reveals distinct failure modes: (a) often ignores auxiliary-line construction and overfits to answer-only cues; (b) produces fluent but visually inconsistent descriptions (e.g., incorrect lines) that fail to constrain diagram-based reasoning. Overall, text-only alignment is intrinsically lossy for geometric structure. Robust auxiliary-line reasoning requires visually grounded, structure-preserving diagram-text alignment that enables rigorous checks of metric accuracy and incidence relations.

**Reinforcement Learning.** To quantify the contribution of reinforcement learning, we ablate it and train the SFT-only variant. As shown in Table 2, GeoVLMath-7B with SFT+RL consistently outperforms the SFT-only model, with pronounced drops once RL is removed. This improvement reflects how RL drives the policy beyond strict imitation: reward-aligned optimization not only encourages exploration of more effective strategies but also provides credit assignment for beneficial intermediate steps, rather than confining learning to surface-level matching. As a result, the model becomes less dependent on dataset-specific heuristics and exhibits more stable performance across the evaluated benchmarks. From this perspective, RL functions as a post-SFT catalyst that unlocks latent capacities of the base model and consolidates preliminary SFT competence into robust multistep reasoning, particularly in scenarios demanding deliberate auxiliary-line construction.

## 5 RELATED WORK

Recent LVLMs (Anthropic, 2025b; DeepMind, 2025a; OpenAI, 2025) have advanced geometric problem solving, especially in plane geometry. The prior methods mainly follow two categories: (i) direct generation of answers or reasoning paths from multimodal inputs (Ning et al., 2025; Xia et al., 2025; Gao et al., 2025), which is limited by intrinsic reasoning capacity; and (ii) tool-augmented reasoning that produces executable code for symbolic computation (Zhao et al., 2025; Sharma et al., 2025; Chen et al., 2024), offloading difficult steps to external engines. However, most methods still handle visual and textual modalities largely separately and lack mechanisms to incorporate auxiliary lines, which restricts performance on tasks requiring such constructions. Visual Sketchpad (Hu et al., 2024) prompts LVLMs to generate code for auxiliary lines that are rendered into augmented diagrams, but relies on precise code and exact coordinate annotations that are rare in real diagrams. In contrast, our approach integrates auxiliary-line construction into a RL framework guided by a cross-modal reward model, decouples construction from the LVLM's intrinsic reasoning, generalizes across model scales, and removes the need for explicit coordinates, enabling robust auxiliary-line reasoning in solid geometry. Details on benchmarks and datasets are provided in Appendix F.

## 6 CONCLUSION

We present a framework for solving solid geometry problems that require auxiliary lines, an underexplored area in vision-language reasoning. To support this task, we curate **AuxSolidMath**, a high-quality dataset from real high-school exams with paired diagrams, problem statements, auxiliary-line descriptions, and final answers, enabling both answer supervision and vision-language reward modeling. At the core is a vision-based reward model that scores the agreement between generated auxiliary-line descriptions and ground-truth auxiliary-line-annotated diagrams, providing stable signals for reinforcement learning. Using this dataset and reward model, we train **GeoVLMath**, an open-source LVLM optimized with RL. Experiments demonstrate state-of-the-art results among open-source LVLMs of comparable size and competitive performance against advanced closed-source LVLMs such as GPT-4o. We will make AuxSolidMath and GeoVLMath publicly available on Hugging Face to facilitate reproducibility, benchmarking, and further research.

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

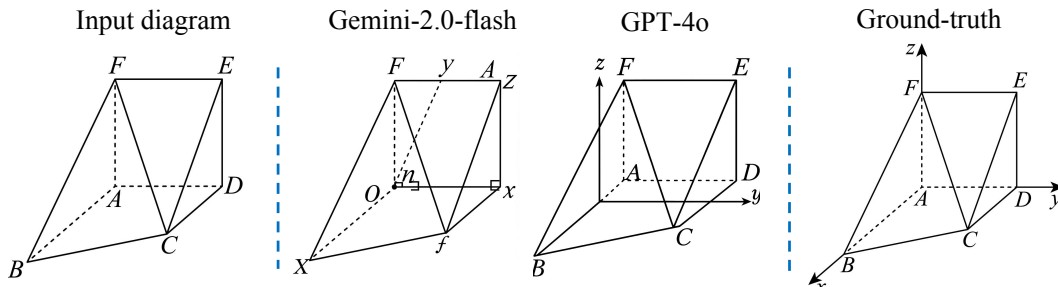

**[Auxliary Line Construction]** Take A as the origin, and let the lines along AB, AD, and AF be the x-axis y-axis, and z-axis, respectively, then establish the three-dimensional Cartesian coordinate system A-xyz.

Figure 5: Comparison of two representative image editing models for constructing a three-dimensional Cartesian coordinate system.

# A    THE USE OF LARGE LANGUAGE MODELS

In this paper, the authors used ChatGPT solely for language polishing, including grammar, phrasing, and stylistic refinement. We did not use it to generate scientific content, such as research ideas, methods, or related work. We did not provide any confidential, personal, or proprietary data to the model. The authors take full responsibility for all scientific content, which was exclusively written and verified by us.

# B    LIMITATION AND FUTURE WORK

Although our framework achieves competitive performance, it does not yet realize the ideal approach of directly rendering precise auxiliary lines on the diagram. Given the limited geometric controllability of current image editing and diffusion models, we instead employ a cross-modal reward model as a practical proxy to promote diagram-text consistency without directly editing the diagram images. In future work, we will investigate constraint-guided diffusion in conjunction with a geometry-constrained rendering engine to explicitly render auxiliary lines, thereby enhancing the alignment between visual constructions and symbolic reasoning.

# C    QUALITATIVE COMPARISON OF IMAGE EDITING MODELS FOR AUXILIARY-LINE GENERATION

To further highlight the limitations of current image-editing models in precise geometric construction, we present a single illustrative example comparing two representative models on a three-dimensional coordinate system construction task. As shown in Figure 5, this comparative example reveals a persistent difficulty in faithfully instantiating the specified auxiliary line descriptions, namely, aligning the edits with the intended spatial constraints, which motivates our text-driven auxiliary line construction guided by a cross-modal reward model.

# D    DATA CREATION

In this section, we detail the four progressive steps of our data creation pipeline.

## D.1    AUTOMATED PROBLEM IDENTIFICATION

To construct the AuxSolidMath dataset, we first collect over 350 sets of high school geometry problems from publicly available online sources. Given that the dataset is intended to support constructive geometric reasoning, we specifically target problems that necessitate auxiliary line constructions as integral components of their solutions.

To efficiently identify such problems, we design an automated two-stage filtering pipeline using Python scripts. In the first stage, we detect problems whose solutions contain explicit mentions of auxiliary-line constructions. Specifically, we apply regular expression patterns to locate question number markers that are explicitly present in the exam papers and use these markers to segment the content into individual problem units. For each problem, we examine the solution for verbs that signal the introduction of auxiliary lines (e.g., "connect," "construct," "draw," "establish"). Problems lacking such terms are discarded, while those containing relevant cues are retained. In the second stage, we further refine the selection by ensuring that each retained problem contains both the original diagram and an auxiliary-line diagram. To this end, we quantify the number of diagrams associated with each problem. Problems with fewer than two diagrams are excluded, whereas those with at least two, which usually represent the original and modified diagrams, are preserved. This automated pipeline enables scalable and consistent filtering of auxiliary-line geometry problems, significantly reducing manual annotation effort.

## D.2 Automated Deduplication and Diagram Extraction

Upon identifying geometry problems requiring auxiliary lines, we employ an automated pipeline to deduplicate instances and extract the associated diagrams. This step guarantees the uniqueness and visual quality of data instances for downstream model training.

**Problem Deduplication.** To eliminate duplicate problems, we retain only the first occurrence of each unique problem based on its textual content. Concretely, we initialize a global problem set as an empty collection. We then sequentially process all Word-formatted exam papers, examining only the problem descriptions while ignoring the associated solutions and diagrams. For each problem, if its description is not already present in the global set, we add the problem; otherwise, we discard it as a duplicate. This procedure ensures that identical problems, which often recur across different examinations, are retained only once.

**Diagram Extraction.** Following deduplication, we extract, filter, and store the geometric diagrams associated with each retained problem. A key challenge lies in reliably distinguishing true geometric figures from image-embedded mathematical expressions (*e.g.*, MathType equations), as both appear in Word exam papers. Existing Python libraries are unable to make this distinction accurately, often misclassifying equations as diagrams and introducing significant noise into the extraction process. To overcome this limitation, we innovatively integrate the Apache POI library through a custom Java implementation, enabling fine-grained control over the parsing of Word documents. This setup enables reliable identification and extraction of genuine geometric diagrams while effectively filtering out formula-rendered images. To further ensure visual quality, the extracted diagrams are then processed using OpenCV to discard low-resolution or unclear diagrams. The remaining diagrams are subsequently saved using a standardized naming convention that distinguishes between the original and the annotated versions of the auxiliary lines. To be more specific, for each problem indexed by i, we store two images: {i}.png, which contains the original diagram, and {i}_auxiliary.png, which includes the corresponding auxiliary-line diagram. This consistent format facilitates downstream alignment between textual and visual modalities within the multimodal processing pipeline.

## D.3 Structured Data Extraction

Building on the high-quality geometric diagrams obtained in the previous step, we proceed to extract the corresponding textual content for each geometry instance, including the problem description, the final answer, and the auxiliary-line description. This extraction process is non-trivial, as the original Word documents frequently embed mathematical expressions using MathType formats that are not reliably handled by standard document parsing tools.

To address this challenge, we render the processed Word documents as images, thereby enabling LVLMs to leverage their visual reasoning capabilities. Although this approach appears straightforward, open-source models such as Qwen2.5-VL-7B-Instruct (Bai et al., 2025) often struggle to accurately parse complex geometry problems involving symbolic notation and mathematical expressions. In contrast, closed-source models like GPT-4o (Hurst et al., 2024) exhibit significantly stronger performance, but their reliance on commercial APIs introduces substantial costs and limits scalability in large-scale applications. To balance accuracy with scalability, we adopt a hybrid strategy. More concretely, we first utilize an advanced closed-source model (*i.e.*, GPT-4o) to gen-

erate a small, high-quality seed dataset comprising 300 manually verified instances. This curated dataset is then used to fine-tune an open-source LVLM (*i.e.*, Qwen2.5-VL-7B-Instruct), resulting in a lightweight, domain-adapted model capable of accurate and scalable text extraction. The final output consists of the extracted problem description, the final answer, and the auxiliary-line description, all encapsulated in a structured JSON format. This unified representation facilitates consistent data handling and serves as a foundation for training a robust open-source text extraction model. By releasing this model, we aim to contribute a practical and reusable resource to the broader research community working on geometry-aware vision-language understanding.

### D.4 Manual Verification

To ensure the quality and reliability of the final dataset, we perform a manual verification step that assesses each data instance in terms of accuracy, completeness, uniqueness, and semantic consistency, alongside visual quality criteria such as image clarity and resolution. Two AI researchers serve as independent checkers. Each instance is independently reviewed by both researchers. If either checker identifies a potential issue, the instance is collaboratively revised. This process is repeated up to three times per instance, ensuring that all errors, including subtle or ambiguous ones, are systematically identified and corrected. Manual verification plays a critical role in resolving complex symbolic expressions and ambiguous diagrammatic content that automated tools may misinterpret. Despite its relatively low cost and effort, this step remains indispensable for ensuring the high data fidelity necessary for a reliable model.

## E Experimental Setup

### E.1 Models

On the closed-source models, we include leading models such as gpt-5-mini (OpenAI, 2025), o4-mini (OpenAI, 2025) and GPT-4o (Hurst et al., 2024), Gemini-2.0-Flash and Gemini-2.5-Flash (DeepMind, 2025a), Claude 3.7 Sonnet (Anthropic, 2025a) and Claude Sonnet 4 20250514 (Anthropic, 2025b). These models represent the forefront of multimodal reasoning among closed-source models, although their internal architectures remain undisclosed. On the open-source models, we consider several publicly available high-performance models, including the Qwen2 VL (Wang et al., 2024b) and Qwen2.5 VL series (Bai et al., 2025), InternVL 3 families (Zhu et al., 2025), LLaMA-3.2-11B-Vision-Instruct (Meta, 2024) and Llama-4-Scout-17B-16E-Instruct (Meta, 2025). These models encompass a range of design paradigms, parameter scales, and instruction tuning strategies, providing a robust comparative foundation for evaluating multimodal reasoning capabilities. Note that models such as Gemini-2.5 Pro (DeepMind, 2025b) and OpenAI o3 (OpenAI, 2025) are excluded from our study due to limited accessibility and high inference costs.

### E.2 Training Implementation Details

We adopt a two-stage training paradigm based on the Qwen2.5-VL series, including Qwen2.5-VL-3B and Qwen2.5-VL-7B, consisting of the SFT stage and the RL stage.

**SFT Stage.** The SFT phase is conducted using the `LLaMA-Factory` framework (Zheng et al., 2024). For Qwen2.5-VL-7B, we train the model for 5 epochs with a per-device batch size of 2 and a gradient accumulation step of 8 (effective batch size of 16). We use the AdamW optimizer with a learning rate of 2e-5 and apply a cosine learning rate scheduler with a warmup ratio of 0.1. The model is trained in `bf16` precision. Vision and projection modules are frozen during this stage, while the language model remains unfrozen. For Qwen2.5-VL-3B, we adopt the same training configuration as the 7B variant, except learning rate and training epochs. Specifically, Qwen2.5-VL-3B is trained for 5 epochs with a learning rate of 3e-5.

**RL Stage.** The RL phase is performed using the `EasyR1` framework (Zheng et al., 2025) with the GRPO algorithm. For the Qwen2.5-VL-7B model, training and validation data are loaded from Parquet files containing question-diagram pairs, with a maximum response length of 8192. Both rollout and validation batch size are set to 16. The actor is optimized using AdamW (learning rate 2e-6, weight decay 1e-2, no warmup). KL regularization is applied using the `low_var_kl`

penalty with a coefficient of 1e-2. Training runs for 6 epochs using `bf16` precision, with gradient checkpointing and partial FSDP offloading enabled for memory efficiency.

*Rewards.* The overall reward is the sum of a cross-modal auxiliary-line consistency reward and a final-answer accuracy reward, where the auxiliary-line component is weighted by $\alpha = 0.1$. For Qwen2.5-VL-3B, we adopt the same RL configuration as the 7B model, with adjustments to the batch size and the number of training epochs. Specifically, Qwen2.5-VL-3B is trained for 4 epochs with a batch size of 8.

All training was conducted on a server equipped with two NVIDIA A100 80GB and two NVIDIA A800 80GB GPUs. The SFT stage was performed on the A100 GPUs, while the full set of four GPUs was utilized during the reinforcement learning stage.

## F RELATED WORK

In this section, we also review benchmarks and datasets for geometric reasoning. Most benchmarks and datasets for geometric problem solving focus on plane geometry, where diagrams and problems involve two-dimensional figures. Well-known resources in this area include Geometry3K (Lu et al., 2021), GeoQA (Chen et al., 2021a), UniGeo (Chen et al., 2022), and GeomRel (Wang et al., 2025b), which primarily cover plane geometry problems. A concurrent benchmark, GeoLaux (Fu et al., 2025), explores the use of auxiliary lines in plane geometry, but is limited to simple cases and lacks engagement with the spatial complexity of solid geometry. Nonetheless, there remains a lack of dedicated resources for solid geometry, even though solving such problems often requires interpreting three-dimensional relationships and drawing auxiliary lines to uncover hidden spatial structures. While SolidGeo (Wang et al., 2025a) is a recent benchmark that focuses exclusively on solid geometry, it does not explicitly require auxiliary lines for solving its problems, leaving this important aspect of spatial reasoning underexplored. Similarly, other benchmarks such as Math-Vista (Lu et al., 2024), MathVision(Wang et al., 2024a), and MathVerse (Zhang et al., 2024) contain only a limited number of solid geometry problems, and these also do not require auxiliary lines to reach the solution. As a result, these resources fall short of evaluating a model's ability to solve complex solid geometry problems where auxiliary lines are essential for uncovering implicit spatial relationships. To address this gap, we present AuxSolidMath, the first dedicated dataset for solid geometry problems that require auxiliary lines to solve. It offers comprehensive multimodal supervision, including the original diagram, the problem statement, textual descriptions of the required auxiliary lines, the final answer, and a corresponding diagram annotated with those lines, enabling models to learn how the auxiliary lines facilitate solid geometry reasoning.

## G PROMPTS

### G.1 PROMPTS FOR SUPERVISED FINE-TUNING

The following presents the two-part prompt template used in our supervised dataset. The system prompt assigns the solver role and enforces formatting: auxiliary lines must be wrapped in `[AUX]...[/AUX]` and the final answer must appear as plain text in `Final Answer:...`. The user prompt is multimodal, pairing a diagram referenced by the `<image>` token with the natural language question `{question}`, which yields explicit reasoning steps and a final answer.

---
**SYSTEM_PROMPT_FOR_SFT**

```
SYSTEM_PROMPT_FOR_SFT = """
```
You are a mathematician skilled in solving geometry problems through step-by-step reasoning. Solve the given geometry problem based on a geometric diagram and a natural language question. Use '[AUX]...[/AUX]' to indicate auxiliary constructions, such as establishing coordinate systems or constructing auxiliary lines. Finally, provide your final answer within 'Final Answer:...'.
```
"""
```

---

**USER_PROMPT_FOR_SFT**

```
USER_PROMPT_FOR_SFT = """
Image: <image>
Question: {question}
"""
```

## G.2 PROMPTS FOR CROSS-MODAL REWARD MODEL

Using the prompt below, the cross-modal reward model compares the description of the auxiliary line generated by the policy model against a pair of diagrams, the original image $I$ and its auxiliary-line counterpart $I^+$, and returns a single line justification and a calibrated score in $[0, 1]$ that measures visual–textual agreement. The instruction emphasizes the correctness of auxiliary-line constructions and adherence to geometric constraints. Higher scores indicate stronger alignment.

**SYSTEM_PROMPT_FOR_CROSS-MODAL_REWARD_MODEL**

```
SYSTEM_PROMPT_FOR_CROSS-MODAL_REWARD_MODEL = """
```
You are a professional geometry reasoning evaluator. Your task is to evaluate whether a given textual description of auxiliary lines accurately explains the visual difference between the original diagram and the auxiliary-line diagram.

Score the description on a scale from 0 to 1:
1. 1 indicates a fully accurate and helpful description.
2. 0 indicates a completely irrelevant or misleading description.
3. Intermediate values (e.g., 0.25/ 0.50/ 0.75) reflect partial relevance or minor issues.

Return exactly one line:
<brief justification>. Score: .
```
"""
```

**USER_PROMPT_FOR_CROSS-MODAL_REWARD_MODEL**

```
USER_PROMPT_FOR_CROSS-MODAL_REWARD_MODEL = """
```
Image (original diagram): <image $I$>
Image (auxiliary-line diagram): <image $I^+$>
Auxiliary-line description: {generated_aux_description}
```
"""
```

# H REPRESENTATIVE EXAMPLES

As illustrad in Figure 6 present qualitative examples from the AuxSolidMath dataset, including the question, the final answer, the auxiliary-line description, the original diagram, and the auxiliary-line diagram. The examples showcase diverse strategies for constructing auxiliary lines and demonstrate that explicit annotations reveal the key spatial constraints.

**Question:** As shown in the diagram, consider the regular truncated quadrilateral pyramid ABCD - $A_1$ $B_1$ $C_1$ $D_1$. Given that $\angle B_1$ BA = $60°$, AB = $2A_1$ $B_1$ = 4, and E is the midpoint of segment CD, determine the sine of the angle between line $ED_1$ and plane $ABB_1$ $A_1$.

**Final Answer:** $\sqrt{6}/3$

Original Diagram    Auxiliary-line Diagram

**Auxiliary-line Description:** Connect $A_1$ $C_1$, which intersects $B_1$ $D_1$ at $O_1$, then connect $OO_1$. Take O as the origin, and let the lines along OB, OC, and $OO_1$ be the x-axis, y-axis, and z-axis, respectively, then establish the three-dimensional Cartesian coordinate system O-xyz.

**Question:** As shown in the diagram, consider the regular truncated quadrilateral pyramid ABCD - $A_1$ $B_1$ $C_1$ $D_1$. The edge lengths of the upper and lower bases are 2 and 3, respectively, and the lateral edge length is $\sqrt{2}$. By extending $AA_1$, $BB_1$, $CC_1$, $DD_1$, they intersect at point P. Determine the volume of the quadrilateral pyramid P - $A_1$ $B_1$ $C_1$ $D_1$.

**Final Answer:** $4\sqrt{6}/3$

Original Diagram    Auxiliary-line Diagram

**Auxiliary-line Description:** Through point P, construct PO perpendicular to the base ABCD at point O, intersecting the plane $A_1$ $B_1$ $C_1$ $D_1$ at point $O_1$. Through point $B_1$, construct $B_1$ E perpendicular to the base ABCD at point E.

**Question:** Given the geometric solid shown in the diagram, the base ABC is an equilateral triangle with a side length of 4. The lateral face $AA_1$ $C_1$ C is a square, and the plane $AA_1$ $C_1$ C is perpendicular to the plane ABC. Point D lies on edge $CC_1$, with CD = (1/4) * $CC_1$, and $BB_1$ = 3CD. Determine the sine of the angle between line B1D and plane $AA_1$ $C_1$ C.

**Final Answer:** $\sqrt{15} / 5$

Original Diagram    Auxiliary-line Diagram

**Auxiliary-line Description:** Let M be the midpoint of AC and N be the midpoint of $A_1$ $C_1$. Then connect MN, BM, and $NB_1$. Through point $B_1$, construct $B_1$ H perpendicular to MN, intersecting MN at point H, then connect DH.

**Question:** As shown in the diagram, consider the quadrilateral pyramid P - ABCD, where the base ABCD is a parallelogram. Points M and N are located on segments PC and PB, respectively. Given that PM:MC = 3:1 and AN $\parallel$ plane BDM, determine the ratio PN:NB.

**Final Answer:** 2

Original Diagram    Auxiliary-line Diagram

**Auxiliary-line Description:** Connect AC, which intersects BD at point O, connect CN, which intersects BM at point G, and then connect OG. Construct NH parallel to BM, intersecting PC at point H.

**Question:** As shown in the diagram, in a pentahedron with vertices A, B, C, D, E, F, quadrilaterals ABCD and ADEF are both isosceles trapezoids. It is given that EF is parallel to AD, BC is parallel to AD, AD = 4, AB = BC = EF = 2, ED = $\sqrt{10}$, FB = $2\sqrt{3}$, and M is the midpoint of AD. Find the sine of the dihedral angle F-BM-E.

**Final Answer:** $4\sqrt{3}/13$

Original Diagram    Auxiliary-line Diagram

**Auxiliary-line Description:** Draw BO perpendicular to AD intersecting AD at O, and connect OF. Take point O as the origin, and let the lines OB, OD, and OF be the x-axis, y-axis, and z-axis, respectively, to establish the three-dimensional Cartesian coordinate system O - xyz.

Figure 6: Representative examples from the AuxSolidMath dataset.

# I CASE STUDY

## I.1 SUCCESSFUL CASES

In this section, we present representative cases produced by GeoVLMath-7B (see Figure 7). The examples illustrate how valid auxiliary-line constructions capture essential spatial constraints, which in turn lead to correct final answers. Collectively, these results provide concrete evidence of both the effectiveness of our training framework and the model's capacity to tackle complex geometric reasoning requiring robust visual understanding and precise symbolic inference.

---

**Question:** As shown in the diagram, in the regular triangular prism $ABC – A_1 B_1 C_1$, the side length of the base is 2, and the lateral edge length is $\sqrt{3}$. D is the midpoint of AC. Find the volume of the triangular pyramid $D - A_1 C_1 B$.

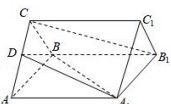

**Solution:** \n**[AUX]1. Construct auxiliary lines: Take the midpoint E of $BC_1$, and connect $A_1$ E and $D_1$ E. [/AUX]**\n2. Determine parallel lines: Since $ABC$-$A_1 B_1 C_1$ is a regular triangular prism, AB $\parallel$ $C_1 D_1$ and AB = $C_1 D_1$. So quadrilateral $ABC_1 D_1$ is a parallelogram.\n3. Identify equal lengths: Given the side length of the base is 2 and the lateral edge length is $\sqrt{3}$, triangle $C_1$ DC is equilateral with side length 2.\n4. Calculate the volume: Since $ABC$–$A_1 B_1 C_1$ is a regular triangular prism and triangle ABC is equilateral with side length 2, the height of the pyramid from D to the plane $A_1 C_1$ B can be calculated as $\sqrt{3}$. The area of the triangular base $A_1 C_1$ B is $(\sqrt{3})/4 * (2)^2 = \sqrt{3}$. Therefore, the volume of the pyramid is $(1/3) *$ base area * height = $(1/3) * \sqrt{3} * \sqrt{3} = 1$.\n\n**Final Answer: 1**.

---

**Question:** As shown in the diagram, in the quadrilateral pyramid S - ABCD, SD is perpendicular to the plane ABCD. The base ABCD is a rhombus, with E and F being the midpoints of SB and AD, respectively. If $\angle BAD = 60°$, SD = 4, and AB = 2, find the volume of the triangular pyramid C - DEF.

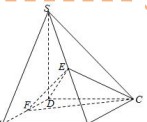

**Solution:** \n **[AUX]1. Construct auxiliary lines: Take the midpoint H of AB, and connect DH. [/AUX]**\n 2. Since $\angle BAD = 60°$ and AB = AD = 2, triangle ABD is equilateral.\n 3. Since H is the midpoint of AB, AH = 1. By Pythagoras theorem in right triangle ADH, DH = $\sqrt{(AD^2 - AH^2)} = \sqrt{(2^2 - 1^2)} = \sqrt{3}$.\n 4. Since ABCD is a rhombus with AB = BC = CD = DA = 2 and $\angle ADC = 120°$, we can calculate DC. Using cosine rule in triangle ADC, $DC^2 = AD^2 + AC^2 - 2*AD*AC*\cos(120°) = 2^2 + (2*2) - 2*2*2*\cos(\pi/3) = 4 + 4 - 4 = 1$. So DC = 1.\n 5. Triangle CDE has vertices: C, D, E. E is the midpoint of SB. Since S is the apex and SD $\perp$ plane ABCD, plane SAB $\perp$ plane ABCD. Therefore, DE $\perp$ plane ABCD. Since CH lies in plane ABCD, we conclude DE $\perp$ CH.\n 6. Since AB = BC = CD = DA = 2, and H is the midpoint of AB, AC $\perp$ BH. Similarly, since CD $\perp$ DA (because $AD^2 + CD^2 = 2^2 + 1^2 = 5$, $AC^2 = 2^2 - 1^2 = 3$) and DC = 1, we can deduce CH $\perp$ CD. Thus CH $\perp$ plane CDE.\n 7. The area of triangle CDE is:\n   Area(CDE) = $(1/2) * DC * DE = (1/2) * 1 * 2 = 1$.\n 8. The height of the pyramid C-DEF is CH. We need to find CH. In triangle CDB, DB = $2\sqrt{2}$ (by Pythagoras theorem). So CH = AC $\sin(60°)$ = 2 * ($\sqrt{3} / 2$) = $\sqrt{3}$.\n 9. The volume of pyramid C-DEF is:\n   Volume(C-DEF) = $(1/3) * $ Area(CDE) * CH = $(1/3) * 1 * \sqrt{3} = \sqrt{3} / 3$.\n\n **Final Answer: $\sqrt{3}/3$**.

---

**Question:** As shown in the diagram, the diameter of the base of the cylinder is 3, and the slant height is 4. AB and CD are the diameters of the upper and lower bases of the cylinder, respectively, and AB $\perp$ CD. Find the volume of the triangular pyramid A-BCD.

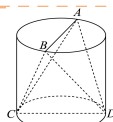

**Solution:** \n **[AUX]1. Construct auxiliary lines: Take the midpoint M of AB, and connect MD and MC.[/AUX]**\n2. Determine the base area of the triangular pyramid: Since AB $\perp$ CD and AB = CD = 3 (diameter), the area of the triangular base ABCD is given by:\n   Area = $(1/2) * AB * CD = (1/2) * 3 * 3 = 4.5$.\n3. Determine the height of the pyramid: The height of the triangular pyramid A-BCD is the perpendicular distance from A to the plane BDC. Given that AB $\perp$ plane BDC, the height is AB itself, which is given as 4 (slant height).\n4. Calculate the volume of the triangular pyramid A-BCD:\n   Volume = $(1/3) *$ Area of base * Height = $(1/3) * 4.5 * 4 = 6$\n\n**Final Answer: 6**.

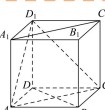

**Question:** As shown in the diagram, the edge length of the cube ABCD – $A_1$ $B_1$ $C_1$ $D_1$ is 2. Find the sine of the angle between line BD and plane $ACD_1$ .

**Solution:** \n[AUX]1. Establish coordinate system: Take point D as the origin, and let the lines DA, DC, and $DD_1$ be the x-axis, y-axis, and z-axis, respectively, to establish the three-dimensional Cartesian coordinate system D-xyz. [/AUX]\n2. Determine coordinates of points: Since the edge length of the cube ABCD – $A_1$ $B_1$ $C_1$ $D_1$ is 2, we have the coordinates of the vertices as follows:\n D = (0, 0, 0)\n A = (2, 0, 0)\n C = (0, 2, 0)\n B = (2, 2, 0)\n $D_1$ = (0, 0, 2)\n $A_1$ = (2, 0, 2)\n $C_1$ = (0, 2, 2)\n $B_1$ = (2, 2, 2)\n3. Find the coordinates of point E: E is the midpoint of BD. The coordinates of B are (2, 2, 0) and D are (0, 0, 0). The midpoint E is the average of the coordinates of B and D:\n E = ((2+0)/2, (2+0)/2, (0+0)/2) = (1, 1, 0).\n4. Find the vector BD: We have point B = (2, 2, 0) and point D = (0, 0, 0).\n Vector BD = D - B = (0 - 2, 0 - 2, 0 - 0) = (-2, -2, 0).\n5. Find a normal vector to the plane $ACD_1$ : To find the normal vector, we need two vectors in the plane $ACD_1$ .\n Vector AC = C - A = (0 - 2, 2 - 0, 0 - 0) = (-2, 2, 0).\n Vector $AD_1$ = $D_1$ - A = (0 - 2, 0 - 0, 2 - 0) = (-2, 0, 2).\nNormal vector n to the plane $ACD_1$ is the cross product of AC and $AD_1$ :\n n = AC × $AD_1$ = (-2, 2, 0) × (-2, 0, 2) = (2*2 - 0*0, 0*(-2) - (-2)*2, (-2)*0 - 2*(-2)) = (4, 4, 4).\nWe can take a simpler normal vector by dividing by 4, so we can take n = (1, 1, 1).\n6. Calculate the sine of the angle between line BD and plane $ACD_1$ : The sine of the angle θ between vector BD and plane $ACD_1$ is given by the absolute value of the cosine of the angle between vector BD and normal vector n.\n sin(θ) = |cos(angle between BD and n)| = |(BD · n) / (|BD| * |n|)|.\n BD · n = (-2, -2, 0) · (1, 1, 1) = (-2)(1) + (-2)(1) + (0)(1) = -2 - 2 + 0 = -4.\n |BD| = √((-2)² + (-2)² + 0²) = √(4 + 4) = √8 = 2√2.\n |n| = √(1² + 1² + 1²) = √(1 + 1 + 1) = √3.\n sin(θ) = |-4 / (2√2 * √3)| = |-4 / (2√6)| = |-2/√6| = 2/√6 = 2√6 / 6 = √6 / 3.\n\n**Final Answer: √6/3.**

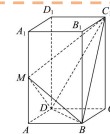

**Question:** As shown in the diagram, in the regular quadrilateral prism ABCD−$A_1$ $B_1$ $C_1$ $D_1$ , M is the midpoint of $AA_1$ , AB = 2, $AA_1$ = 4. Find the cosine value of the dihedral angle $C_1$ -BD-M.

**Solution:** \n[AUX]1. Establish coordinate system: Take point D as the origin, and let the lines DA, DC, and $DD_1$ be the x-axis, y-axis, and z-axis, respectively, to establish the three-dimensional Cartesian coordinate system D-xyz. [/AUX]\n2. Determine coordinates of points: Since ABCD–$A_1$ $B_1$ $C_1$ $D_1$ is a regular quadrilateral prism with base side length AB = 2 and height $AA_1$ = 4, we have the coordinates of the vertices as follows:\n D = (0, 0, 0)\n A = (2, 0, 0)\n C = (0, 2, 0)\n B = (2, 2, 0)\n $D_1$ = (0, 0, 4)\n $C_1$ = (0, 2, 4)\n $A_1$ = (2, 0, 4)\n M is the midpoint of $AA_1$ , so M = ((2+2)/2, (0+0)/2, (0+4)/2) = (2, 0, 2).\n3. Find normal vectors of planes $C_1$ BD and MBD:\n For plane $C_1$ BD, we need two vectors in the plane: $C_1$ D and $C_1$ B.\n Vector $C_1$ D = D - $C_1$ = (0, 0, 0) - (0, 2, 4) = (0, -2, -4).\n Vector $C_1$ B = B - $C_1$ = (2, 2, 0) - (0, 2, 4) = (2, 0, -4).\n The normal vector $n_1$ of plane $C_1$ BD is the cross product of $C_1$ D and $C_1$ B:\n $n_1$ = $C_1$ D × $C_1$ B = (0, -2, -4) × (2, 0, -4) = ((-2)(-4) - (-4)(0), (-4)(2) - 0*(-4), 0(0) - (-2)(2)) = (8, -8, 4).\n We can take a simpler normal vector by dividing by 4, so $n_1$ ' = (2, -2, 1).\n For plane MBD, we need two vectors in the plane: MB and MD.\n Vector MB = B - M = (2, 2, 0) - (2, 0, 2) = (0, 2, -2).\n Vector MD = D - M = (0, 0, 0) - (2, 0, 2) = (-2, 0, -2).\n The normal vector $n_2$ of plane MBD is the cross product of MB and MD:\n $n_2$ = MB × MD = (0, 2, -2) × (-2, 0, -2) = (2(-2) - (-2)(0), (-2)(-2) - 0(-2), 0(0) - 2(-2)) = (-4, 4, 4).\n We can take a simpler normal vector by dividing by 4, so $n_2$ ' = (-1, 1, 1).\n4. Calculate the cosine of the dihedral angle:\n The cosine of the dihedral angle θ is the cosine of the angle between the normal vectors $n_1$ ' and $n_2$ ':\n cos(θ) = |($n_1$ ' · $n_2$ ') / (|$n_1$ '| * |$n_2$ '|)|.\n $n_1$ ' · $n_2$ ' = (2, -2, 1) · (-1, 1, 1) = 2(-1) + (-2)(1) + 1(1) = -2 - 2 + 1 = -3.\n |$n_1$ '| = √(2² + (-2)² + 1²) = √(4 + 4 + 1) = √9 = 3.\n |$n_2$ '| = √((-1)² + 1² + 1²) = √(1 + 1 + 1) = √3.\n cos(θ) = |-3 / (3 * √3)| = |-3 / (3√3)| = |-1/√3| = 1/√3 = √3/3.\n\n**Final Answer: √3/3.**

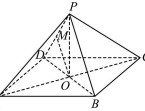

**Question:** In the quadrilateral pyramid P-ABCD, the quadrilateral ABCD is a rhombus, with AC and BD intersecting at point O. PA = PC, PB = PD, ∠BAD = 60 ˚ , AB = 2, and M is the midpoint of segment PD. If the angle between line OM and plane ABCD is 60 ˚ , find the sine of the dihedral angle between plane PAD and plane PBC.

**Solution:** \n[AUX]1. Establish coordinate system: Take point O as the origin, and let the lines OA, OB, and OP be the x-axis, y-axis, and z-axis, respectively, to establish the three-dimensional Cartesian coordinate system O-xyz. [/AUX]\n2. Determine coordinates of points: Since the rhombus ABCD has side length 2 and ∠BAD = 60 ˚ , triangle ABD is equilateral, and triangle BCD is also equilateral. The diagonals AC and BD are perpendicular bisectors of each other. Given AB = 2, we have AO = OC = 1 and BO = OD = √(2^2 - 1^2) = √3. Thus, the coordinates are:\n O = (0, 0, 0)\n A = (1, 0, 0)\n C = (-1, 0, 0)\n B = (0, √3, 0)\n Since PA = PC and PA = sqrt((1-0)^2 + (0-h)^2 + k^2), PC = sqrt((-1-0)^2 + (0-h)^2 + k^2).\n Squaring both PA and PC and equating them:\n (1-h)^2 + k^2 = (-1-h)^2 + k^2\n h = 0\n Substituting h = 0 into PA^2 = (1-0)^2 + (0-0)^2 + k^2 = 1 + k^2 = 1^2 = 1 + (h^2 =0) = PB^2.\n So z coordinate of P = OM = sin(60 ˚ ) = sqrt(3)/2\n M is the midpoint of PD, M = ((0+0)/2, (0+0)/2, sqrt(3)/2 + sqrt(3)/2)/2 = (0, 0, sqrt(3)/2)\n B = (0, sqrt(3), 0)\n P = (0, 0, sqrt(3))\n A = (1, 0, 0), B = (0, √3, 0), C = (-1, 0, 0), D = (0, -√3, 0), P = (0, 0, √3), M = (0, 0, √3/2)\n3. Find normal vectors of plane PAD and plane PBC:\n For plane PAD, we need vectors PA and PD.\n PA = A - P = (1, 0, -√3)\n PD = D - P = (0, -√3, -√3)\n Normal vector n1 of plane PAD is the cross product of PA and PD:\n n1 = PA x PD = (1, 0, -√3) x (0, -√3, -√3) = (-3, √3, -√3)\n For plane PBC, we need vectors PB and PC.\n PB = B - P = (0, √3, -√3)\n PC = C - P = (-1, 0, -√3)\n Normal vector n2 of plane PBC is the cross product of PB and PC:\n n2 = PB x PC = (0, √3, -√3) x (-1, 0, -√3) = (-3, √3, √3)\n4. Calculate the cosine of the dihedral angle:\n The cosine of the dihedral angle θ :\n cos(θ) = |(n1 · n2) / (|n1| * |n2|)|\n n1 · n2 = (-3)(-3) + (√3)(√3) + (-√3)(√3) = 9\n |n1| = √((-3)^2 + (√3)^2 + (-√3)^2) = √15\n |n2| = √((-3)^2 + (√3)^2 + (√3)^2) = √15\n cos(θ) = |9 / (√15 * √15)| = 9 / 15 = 3 / 5\n5. Calculate the sine of the dihedral angle:\n sin^2(θ) = 1 - cos^2(θ) = 16 / 25\n sin(θ) = √(16 / 25) = 4/5 \n**Final Answer: 4/5.**

Figure 7: Successful cases generated by GeoVLMath-7B.

## I.2 FAILURE CASES

In this section, we illustrate typical failure cases generated by GeoVLMath-7B (see Figure 8). The observed failures are exemplified by mis-specified coordinate systems and auxiliary-line descriptions that are irrelevant to the diagram, both of which fail to capture essential spatial constraints and may yield incorrect final answers. To mitigate such errors, we will explore diffusion-based drawing modules that render auxiliary lines directly on the original diagram and support iterative correction as part of future work.

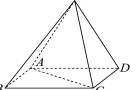

**Question:** As shown in the diagram, in the quadrilateral pyramid P-ABCD, the base ABCD is a square, PAD is an equilateral triangle, and PC = AC. Find the sine of the angle between line PB and plane PCD.

**Ground Truth:** √6/4

**Analysis:** The A–xyz coordinate system is incorrectly established—**AP is not orthogonal to AB or AD**, violating the pairwise orthogonality required for a three-dimensional Cartesian system. Consequently, the coordinates assigned to points and lines are invalid, leading to an incorrect final answer.

**Solution:** \n **[AUX]1. Establish coordinate system: Take point A as the origin, and let the lines AB, AD, and AP be the x-axis, y-axis, and z-axis, respectively, to establish the three-dimensional Cartesian coordinate system A-xyz. [/AUX]**\n2. Determine the coordinates of the vertices: Let the side length of the square ABCD be 2. Since A is the origin, A = (0, 0, 0). As AB is along the x-axis and B is to the right of A with length 2, B = (2, 0, 0). As AD is along the y-axis and D is above A with length 2, D = (0, 2, 0). Since ABCD is a square, C = (2, 2, 0). As PA is a line from A in the z-direction upwards and triangle PAD is an equilateral triangle with side length 2, the height from P to the base ABCD must be √(2^2 - 1^2) = √3. So, P = (0, 0, √3).\n3. Find the vector PB: We have P = (0, 0, √3) and B = (2, 0, 0). The vector PB = B - P = (2, 0, 0) - (0, 0, √3) = (2, 0, -√3).\n4. Find the normal vector of the plane PCD: We need two vectors in the plane PCD, for example, vector PC and vector PD.\nVector PC = C - P = (2, 2, 0) - (0, 0, √3) = (2, 2, -√3).\nVector PD = D - P = (0, 2, 0) - (0, 0, √3) = (0, 2, -√3).\nThe normal vector n of the plane PCD is the cross product of PC and PD:\nn = PC × PD = (2, 2, -√3) × (0, 2, -√3) = (2*(-√3) - (-√3)*2, (-√3)*0 - 2*(-√3), 2*2 - 2*0) = (-2√3 + 2√3, 0 + 2√3, 4 - 0) = (0, 2√3, 4).\nTo take a simpler normal vector, we can use n' = (0, √3, 2).\n5. Calculate the sine of the angle between line PB and plane PCD: The sine of the angle θ between line PB and plane PCD is given by the absolute value of the cosine of the angle between vector PB and normal vector n':\nsin(θ) = |cos(angle between PB and n')| = |(PB · n') / (|PB| * |n'|)|.\nPB · n' = (2, 0, -√3) · (0, √3, 2) = 2*0 + 0*√3 + (-√3)*2 = -2√3.\n|PB| = √(2^2 + 0^2 + (-√3)^2) = √(4 + 0 + 3) = √7.\n|n'| = √(0^2 + (√3)^2 + 2^2) = √(0 + 3 + 4) = √7.\nsin(θ) = |-2√3 / (√7 * √7)| = |-2√3 / 7| = 2√3 / 7.\n\n**Final Answer: 2√3/7.**

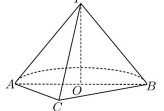

**Question:** As shown in the diagram, the given solid consists of half of a cone and a triangular pyramid. The base of the cone is a circle with center O and radius 1, and the height of the cone satisfies PO = 2. The triangular pyramid P-ABC has its base ABC as an isosceles right triangle, where AB is the hypotenuse and coincides with the diameter of the cone's base circle. The base ABC lies on the same plane as the cone's base. Determine the angle between line PC and plane ABC.

**Ground Truth:** arctan 2

**Analysis:** The generated auxiliary-line description **leaves the geometric configuration unchanged**. Consequently, the subsequent reasoning proceeds from a spurious auxiliary premise rather than from added geometric constraints, leading to an incorrect final answer.

**Solution:** \n**[AUX]1. Construct auxiliary lines: Take the midpoint M of AB, and connect OM. [/AUX]**\n2. Identify perpendicularity: Since O is the center of the circle, OM ⊥ AB. Also, PO ⊥ OM because PO is the height of the cone and O is on the base circle. Since AB ∩ PO = O, and AB and PO are in plane PAB, we conclude OM ⊥ plane PAB.\n3. Determine angle: The angle between line PC and plane ABC is the angle between line PC and its projection onto plane ABC. Since OM ⊥ plane PAB, the projection of point P onto plane ABC lies on line OM. The projection of line PC onto plane ABC is segment AC.\n4. Calculate angle: To find the angle between line PC and plane ABC, we need to find the sine of the angle. Given PO = 2 and OA = 1, we have AC = √(PA² - AO²) = √(2² - 1²) = √3.\n\n**Final Answer: arcsin(√3 / 3)**

Figure 8: Failure cases generated by GeoVLMath-7B.

