# OpenReview forum: "GeoVLMath: Enhancing Geometry Reasoning in Vision-Language Models via Cross-Modal Reward for Auxiliary Line Creation"
_ICLR.cc/2026/Conference — ICLR 2026 Conference Withdrawn Submission_

### Official Review · Reviewer_gPh1 · 2025-10-30

**Soundness:** 2
**Presentation:** 3
**Contribution:** 2
**Rating:** 2
**Confidence:** 3

**Summary:**

This paper addresses the challenge of auxiliary line construction in solid geometry reasoning for vision-language models (LVLMs). Existing methods (e.g., direct image editing, coordinate-dependent tool use) fail to reliably generate precise auxiliary lines due to geometric precision limitations and coordinate scarcity. To solve this, the authors propose GeoVLMath, an open-source LVLM optimized via a two-stage training framework (supervised fine-tuning + GRPO-based reinforcement learning) guided by a cross-modal reward model. This reward evaluates the consistency between text descriptions of auxiliary lines and ground-truth auxiliary-line diagrams, avoiding image manipulation or coordinate reliance. To support training, they construct AuxSolidMath, a dataset of 3,018 real high-school solid geometry problems with paired diagrams, auxiliary-line descriptions, and answers. Experiments show GeoVLMath (3B/7B scales) outperforms larger open-source models (e.g., Qwen2.5-VL-32B) and competes with closed-source models (e.g., GPT-4o) on auxiliary-line reasoning benchmarks.

**Strengths:**

1. For the first time, the task of "auxiliary line construction in solid geometry" is explicitly decomposed into the core problems of "textual description generation" and "cross-modal geometric consistency alignment," bypassing the technical bottlenecks of existing approaches that rely on image editing or precise coordinates, and offering a potentially lightweight solution pathway for geometric reasoning.
2. The constructed AuxSolidMath dataset and GeoAuxBench benchmark fill a critical data gap in the field of auxiliary-line-based solid geometry reasoning, enabling and accelerating follow-up research—such as complex auxiliary line generation and cross-scenario geometric reasoning—and thus demonstrating clear potential to advance the field.

**Weaknesses:**

1. The paper sidesteps the core challenge of "directly and precisely drawing auxiliary lines on the diagram," opting instead for an indirect approach that relies on textual descriptions of such lines. Although this current approach avoids the precision issues inherent in image editing, ambiguity remains in the mapping between textual descriptions and geometric figures. For instance, textual expressions of complex spatial relationships, such as perpendicular skew lines or bisectors of polyhedral angles, can easily become ambiguous. Moreover, the approach heavily depends on the model's precise parsing of natural language, which increases the fragility of the reasoning chain.
2. Although the dataset is divided into "Easy" and "Hard" difficulty levels, it does not explicitly annotate the key factors driving this difficulty—such as the number of auxiliary lines, the number of spatial reasoning steps, or whether cross-plane constructions are required. As a result, performance analysis of models lacks specificity, making it impossible to determine whether a model's weakness lies in constructing multiple auxiliary lines or in reasoning about implicit spatial relationships.
3. When generating textual descriptions of auxiliary lines, should the model prioritize labeling core auxiliary lines—such as those critical to solving the problem, like establishing a coordinate system or connecting midpoints—and then add secondary ones? When auxiliary lines involve coordinated constructions across multiple elements (e.g., "draw a perpendicular from point P to plane ABC, intersecting plane DEF at point Q, then connect QR"), the textual description can become lengthy and hard to parse. This increased complexity may also cause the model to suffer reasoning breakdowns.
4. The experiments rely solely on "Pass@k" (solution accuracy) as the core metric and do not evaluate a critical capability dimension: the precision of auxiliary line descriptions. It would be valuable to supplement this with a quantitative measure of "geometric consistency between the model-generated auxiliary line descriptions and the ground-truth solutions"—for example, by assessing whether key constraints are omitted or incorrect spatial relationships are introduced. Currently, this aspect is only indirectly reflected through reward model scores, lacking interpretable and direct evaluation metrics.

**Questions:**

1. Would the authors be willing to report the inference speed of GeoVLMath (3B/7B parameters), such as the average inference time per problem, and how it compares to other models?
2. Would it be possible to involve domain experts in conducting a practicality evaluation of the model-generated textual descriptions of auxiliary lines？Experts could assess whether the descriptions enable solvers to accurately reconstruct the auxiliary lines and successfully complete the problems.

---

### Official Review · Reviewer_ujvY · 2025-10-31

**Soundness:** 3
**Presentation:** 3
**Contribution:** 2
**Rating:** 4
**Confidence:** 3

**Summary:**

The paper proposes GeoVLMath, a framework designed to enhance large vision-language models (LVLMs) in solving solid geometry problems. The approach leverages an automated data pipeline to construct tuples of original diagram, textual description of auxiliary lines, and diagram with auxiliary lines drawn. Using these triplets, the authors train a reward model, which subsequently guides GRPO-based fine-tuning of the LVLM. To evaluate the effectiveness of the method, the authors introduce GeoAuxBenchmark, a benchmark derived from the automatically collected dataset and tailored to assess reasoning in solid geometry problem solving.

**Strengths:**

- The paper proposes a cross modal reward model that scores diagram text consistency between an auxiliary line description and the ground truth diagram. This is a technically novel contribution that bypasses the limitations of direct visual editing or coordinate based construction, and bridges textual auxiliary line reasoning with geometric structure. This reward formulation directly targets spatial consistency rather than surface level lexical similarity, offering a principled way to supervise geometry aware reasoning.
- The authors build AuxSolidMath, a curated dataset of 3,018 real high school solid geometry problems including problem text, auxiliary line descriptions, diagrams, and verified annotations. The dataset creation pipeline includes automated filtering, diagram extraction, LaTeX parsing of MathType formulas, and manual validation, which strengthens data quality and reproducibility. This fills a real gap in multimodal geometry benchmarks, which have been heavily skewed toward plane geometry.
- The model trained with the cross modal reward shows clear gains over the base models on the proposed benchmark. Ablation results show performance drops when removing the cross modal reward or replacing it with a purely textual objective, validating the necessity of geometry grounded feedback for reliable auxiliary line reasoning.

**Weaknesses:**

While the paper proposes an innovative approach to auxiliary-line reasoning and demonstrates clear performance gains, several limitations remain that constrain the generality and interpretability of the results.

- The evaluation scope is limited to the authors’ own benchmark (*GeoAuxBench*), which is derived from the same dataset used for training (*AuxSolidMath*). This introduces a risk of domain overfitting and makes it difficult to verify whether the proposed approach generalizes to broader geometric reasoning or real-world multimodal reasoning tasks. No experiments are reported on standard public datasets such as MathVista, GeoQA, or MathBench, limiting external validity. For example, it would be necessary to compare the model performance against more general PGPS models such as [1] against standard public dataset to check the influence of the fine-tuning process suggested in the paper.
- The dataset itself remains relatively small (3,018 examples) and homogeneous, as all problems are sourced from similar types of high school solid geometry exams. This narrow distribution may lead the model to learn stylistic or structural regularities rather than genuinely transferable geometric reasoning. A more diverse dataset including different geometry domains, problem types, and diagram styles would strengthen the claim of scalability.
- Ablation studies do not include quantitative analyses on training stability, reward variance, or convergence behavior during GRPO optimization. Given that reinforcement learning is notoriously sensitive to hyperparameters and reward shaping, the robustness and reproducibility of the RL results are uncertain. The absence of multiple training seeds or standard deviation reporting further limits confidence in statistical significance.
- Baseline comparisons may not be entirely fair. Some compared models (e.g., GPT-4o, Qwen2.5-VL-32B) differ greatly in architecture, scale, and tuning objectives. Moreover, the experimental setup does not clarify whether inference parameters such as temperature, context length, or sampling strategies were standardized. These omissions make it difficult to isolate the true contribution of the proposed method relative to scale and decoding variance.
- Although the paper highlights the advantage of textual auxiliary-line construction over image-based approaches, it sidesteps the ultimate goal of visually precise rendering. The method improves diagram-text alignment indirectly through a reward model but does not achieve explicit geometric construction. This reliance on textual intermediates leaves open the question of how well such descriptions correspond to verifiable spatial accuracy in the visual domain.

[1]  Cho, Seunghyuk, et al. "GeoDANO: Geometric VLM with Domain Agnostic Vision Encoder." EMNLP 2025

**Questions:**

- It would be valuable to know how the proposed method performs on broader benchmarks that include solid geometry problems, such as MathVerse or SolidGeo.
- The paper mentions that a Qwen2.5-VL extractor was fine-tuned on only 300 samples to extract text descriptions from the collected problems. The number of samples appears rather limited for robust extraction. Would it not be more effective to employ existing OCR models such as Dots.OCR or DeepSeek-OCR which are known for accurate extraction of mathematical expressions?

---

### Official Review · Reviewer_VoWG · 2025-11-03

**Soundness:** 3
**Presentation:** 3
**Contribution:** 3
**Rating:** 4
**Confidence:** 3

**Summary:**

This paper proposes GeoVLMath, an LVLM with improved performance in solving geometric problems utilizing auxiliary lines. The method proposes utilizing the text modality for auxiliary line generation in LVLM and a novel reward function that scores generated text auxiliary lines with an image diagram for RL fine-tuning.

**Strengths:**

1. The authors propose, AuxSolidMath, a 3K example dataset consisting of a paired geometry diagram and auxiliary lines in textual format for GRPO finetuning.
2. The authors propose a novel reward function that calculates alignment between generated auxiliary lines in text format with the ground-truth image diagram.

**Weaknesses:**

1. While the paper introduces an interesting approach to improving auxiliary line generation, the technical contribution appears limited to the design of the reward model.
2.  The model is significantly behind the closed-source model in terms of performance, including gpt-5-mini, and I believe that the direct comparison with these models is unfair. The proposed method primarily focuses on improving auxiliary line generation and could be further thought of as consisting of two substeps: i) Generating auxiliary lines, and ii) Reasoning and solving the geometrical problem. Can the authors utilize generated auxiliary lines for part i) and utilize more advanced reasoning methods, including GPT-5  for part ii)?

**Questions:**

Overall, the authors have clearly motivated the problem and the need for auxiliary lines in Fig. 1, and the proposed AuxSolidMath dataset is an important step toward improving LVLM performance on geometric problems that utilize auxiliary lines.  However, I am primarily concerned about the limited technical novelty of the proposed method and weak baselines. Please see the weakness for more details.

---

### Note · Authors · 2025-11-12

**Comment:**

We decided to withdraw this submission to make substantial revisions.

**Withdrawal Confirmation:**

I have read and agree with the venue's withdrawal policy on behalf of myself and my co-authors.